# FORMALML: A BENCHMARK FOR EVALUATING FORMAL SUBGOAL COMPLETION IN MACHINE LEARNING THEORY

**Xiao-Wen Yang**[1,2*], **Zihao Zhang**[1*], **Jianuo Cao**[1,3], **Zhi Zhou**[1], **Zenan Li**[4],
**Lan-Zhe Guo**[1,3], **Yuan Yao**[1], **Taolue Chen**[5], **Yu-Feng Li**[1,2†], **Xiaoxing Ma**[1†]

[1]State Key Laboratory of Novel Software Technology, Nanjing University
[2]School of Artificial Intelligence, Nanjing University
[3]School of Intelligence Science and Technology, Nanjing University
[4]Department of Computer Science, ETH Zürich, Switzerland
[5]School of Computing and Mathematical Sciences, Birkbeck, University of London, UK
yangxw@lamda.nju.edu.cn, zihaozhang@smail.nju.edu.cn

## ABSTRACT

Large language models (LLMs) have recently demonstrated remarkable progress in formal theorem proving. Yet their ability to serve as practical assistants for mathematicians, filling in missing steps within complex proofs, remains underexplored. We identify this challenge as the task of *subgoal completion*, where an LLM must discharge short but nontrivial proof obligations left unresolved in a human-provided sketch. To study this problem, we introduce **FormalML**, a Lean 4 benchmark built from foundational theories of machine learning. Using a translation tactic that converts procedural proofs into declarative form, we extract 4,937 problems spanning optimization and probability inequalities, with varying levels of difficulty. FormalML is the first subgoal completion benchmark to combine premise utilization and complex research-level contexts. Evaluation of state-of-the-art provers highlights persistent limitations in accuracy and efficiency, underscoring the need for more capable LLM-based theorem provers for effective subgoal completion.

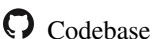 Codebase          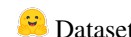 Dataset

## 1 INTRODUCTION

Recent advancements in large language models (LLMs) have showcased their impressive performance in formal theorem proving (Ren et al., 2025; Wang et al., 2025; Xin et al., 2025), particularly culminating in silver-medal level performance at the 2025 International Mathematical Olympiad (Chen et al., 2025). Nevertheless, their effectiveness as copilots for mathematicians in tackling research-level problems (Yang et al., 2024) remains far from satisfactory. For example, the recent Equational Theories Project (Tao et al., 2024) reported that "LLMs did not provide useful suggestions beyond what the human participants could already propose" in most of the difficult cases.

In this paper, rather than attempting to generate a complete proof for such research-level problems, we investigate an intermediate milestone, which we refer to as *subgoal completion*. To elaborate, human experts specify the problem in natural language, formalize the theorem, and outline the high-level proof structure, while AI systems assist by filling in the missing technical details (Jiang et al., 2022). Figure 1 illustrates this process using the proof of gradient descent convergence (Li et al., 2024a; 2025b); The expert provides the informal statement and the main reasoning steps. At the natural language level, the proof appears complete; however, once formalized, it remains incomplete, as indicated by the `sorry` placeholders. This gap arises because the theorem prover cannot automatically bridge each reasoning step. Subgoal completion precisely aims to bridge this gap, aligning the informal and formal proofs.

---

[*]Equal Contribution.
[†]Corresponding Author.

**Theorem.** If $f$ is differentiable, and its gradient $f'$ is $l$-Lipschitz, then the gradient descent with stepsize $a \in [0, \frac{l}{2}]$ at any point $x$ satisfies

$$f(x - af'(x)) \leq f(x) - \frac{a}{2} \| f'(x) \|^2$$

**Proof.** Let's plug in the gradient descent update by letting $x' = x - a\nabla f(x)$. We then get:

$$f(x') \leq f(x) + \langle \nabla f(x), x' - x \rangle + \frac{1}{2} l \|x' - x\|_2^2$$

$$= f(x) + \langle \nabla f(x), x - a\nabla f(x) - x \rangle + \frac{1}{2} l \|x - t\nabla f(x) - x\|_2^2$$

$$\leq f(x) - a\|\nabla f(x)\|_2^2 + \frac{1}{2} l\, a^2 \|\nabla f(x)\|_2^2$$

$$= f(x) - \frac{a}{2}\|\nabla f(x)\|_2^2$$

```
/-- the bound for one step of the gradient method
    using the Lipschitz continuity of the gradient -/

theorem convex_lipschitz
(h₁ : ∀ x₁ : E, HasGradientAt f (f' x₁) x₁)
(h₂ : l > (0 : ℝ)) (ha₁ : l ≤ 1 / a) (ha₂ : a > 0)
(h₃ : LipschitzWith l f') :
∀ x : E, f (x - a • (f' x)) ≤ f x - a / 2 * ‖f' x‖ ^ 2 := by
intro x
calc
  _ ≤ f x + inner (f' x) (x - a • (f' x) - x)
          + l / 2 * ‖x - a • (f' x) - x‖ ^ 2 := by sorry
  _ = f x + ((l.1 / 2 * a * a -a) * ‖f' x‖ ^ 2) := by sorry
  _ ≤ f x + (- a / 2* ‖(f' x)‖ ^2) := by sorry
  _ = f x - a / 2 * ‖f' x‖ ^ 2 := by sorry
```

Figure 1: The informal and formal versions of the statement and proof regarding the sufficient decrease in gradient descent. Both the formal statement and proof can be accurately formalized from the natural language version. However, to verify each step in Lean 4, there are unproven holes (notated as `sorry`) that need to be further completed by existing automated theorem provers.

Despite its natural role as an interface between informal and formal reasoning, subgoal completion still lacks dedicated benchmarks. Existing evaluations are built around competition-style problems and emphasize full-proof generation, whereas our target lies in research-level theorems and the task of completing intermediate proofsteps. Moreover, while proof-generation benchmarks primarily measure the LLM performance by proving success rate, subgoal completion poses a different challenge: handling complex proof contexts efficiently without resorting to verbose or speculative reasoning chains. In practice, a useful prover must therefore strike a balance between accuracy and conciseness. To achieve these goals, we make the following contributions:

- **A Lean 4 translation tactic for subgoal extraction.** We propose a symbolic strategy for the dataset construction. Specifically, we implemented a new Lean 4 tactic that extracts subgoal problems from procedural-style proofs. By adapting the length of the proof segments during extraction, we further utilize this tactic to generate new problems with varying levels of proving difficulty.

- **The FormalML benchmark.** We narrow the scope to the foundational theory of machine learning (ML) and establish a new benchmark in this domain. Our motivation is twofold. First, AI agents are becoming increasingly central to automating scientific discovery (Romera-Paredes et al., 2024), particularly within machine learning research (Lu et al., 2024; Yamada et al., 2025; Gottweis et al., 2025). Hence, automated theorem proving plays a key role in guaranteeing the soundness of derived results. Second, automation tools that verify theoretical correctness can substantially ease the burden on human reviewers (Xu et al.; Pineau et al., 2021), especially given the rapidly growing volume of ML conference submissions. A well-designed benchmark would accelerate the development of such tools. To this end, building on two Lean 4 libraries—Optlib (Li et al., 2024a) and FoML (Sonoda et al., 2025), we establish a dataset of 4,937 subgoal completion problems.

- **Systematic evaluation of LLM-based provers on FormalML**. We evaluate state-of-the-art LLM-based provers on FormalML, highlighting fundamental limitations in accuracy, token-efficiency, and premise utilization. Our results show that while models such as DeepSeek-Prover-V2 (Ren et al., 2025) improve premise utilization capabilities, overall performance drops sharply on higher-difficulty problems. Moreover, chain-of-thought prompting, though effective in natural language reasoning, fails to improve proof completion and often reduces efficiency in this context. Additionally, we find expert iteration to be a effective training approach on FormalML. These insights highlight the need for further development and refinement in LLM-based theorem provers to better support mathematicians in their work.

## 2 RELATED WORK

**LLM-based Theorem Proving.** In recent years, the rapid development of LLMs has significantly advanced research on formal theorem proving (Li et al., 2024b). Current mainstream approaches can be broadly divided into two categories. The first category employs tree-search strategies, including best-first search (Yang et al., 2023; Xin et al., 2025; Ying et al., 2024; Polu & Sutskever, 2020), Monte Carlo tree search (Xin et al., 2024; Wang et al., 2023; Kocsis & Szepesvári, 2006; Yang et al., 2025), and others (Lin et al., 2024). Although these methods align with the Markov Decision Process (MDP) framework of theorem proving, their step-by-step tactic generation and frequent interaction with the Lean environment result in low search efficiency. The second category is whole-

proof generation (Dong & Ma, 2025; Lin et al., 2025a; Zhang et al., 2025; Wang et al., 2025; Ren et al., 2025; Chen et al., 2025), where LLMs directly generate complete proofs, usually through multiple rollouts verified in Lean to identify the final proof. This approach has demonstrated notable advantages. For instance, DeepSeek-Prover-V2 (Ren et al., 2025) and Kimina-Preview (Wang et al., 2025) leverage natural language-aided Long-CoT reasoning to produce formal proofs, achieving breakthroughs in competition-level tasks. However, the performance of existing LLM-based theorem provers in supporting practical formal proving has not yet been fully evaluated.

**Formal Theorem Proving Benchmarks.** The current mainstream Lean formal theorem proving benchmarks can be categorized into three primary types based on problem characteristics. The first type focuses on Olympiad-level high school mathematics competition problems, exemplified by: the MiniF2F (Zheng et al., 2022) dataset, which comprises 244 challenging problems from competitions such as AMC, AIME, and IMO; and the PutnamBench benchmark (Tsoukalas et al., 2024), featuring problems derived from the Putnam Mathematical Competition. The second type targets undergraduate-level mathematical tasks, including ProofNet (Azerbayev et al., 2023) (containing analysis and algebra problems at the undergraduate level) and MinCTX (Hu et al., 2024) (theorems sourced from real-world Lean projects and textbooks). Additionally, hybrid benchmarks like FormalMATH (Yu et al., 2025) cover theorem proving across domains ranging from high school to undergraduate mathematics. Another category, exemplified by Leandojo (Yang et al., 2023), directly constructs datasets from the Mathlib library. Notably, existing benchmarks either operate within simple contexts or do not require premises, and all involve full proof generation tasks. In contrast, our proposed FormalML benchmark specializes in subgoal completion tasks, a design better aligned with practical applications where LLM-based theorem provers assist humans in practical theorem proving. A comprehensive comparative analysis of benchmarks is presented in Table 1.

Table 1: Comparison of existing Lean 4 benchmarks.

| Benchmark | # Problems | Type | Premise | Complex Context | Subgoal Completion |
|---|---|---|---|---|---|
| MiniF2F (Zheng et al., 2022) | 244 | Olympiad | ✗ | ✗ | ✗ |
| PutnamBench (Tsoukalas et al., 2024) | 522 | Olympiad | ✗ | ✗ | ✗ |
| ProofNet (Azerbayev et al., 2023) | 186 | Undergraduate (UG) | ✗ | ✓ | ✗ |
| FormalMATH (Yu et al., 2025) | 5,560 | Olympiad & UG | ✗ | ✓ | ✗ |
| LeanDojo (Yang et al., 2023) | 2000 | Mathlib | ✓ | ✗ | ✗ |
| MiniCTX (Hu et al., 2024) | 762 | UG | ✓ | ✓ | ✗ |
| FormalML (Ours) | 4,937 | UG | ✓ | ✓ | ✓ |

## 3 FORMALML

### 3.1 SUBGOAL EXTRACTION

With the growing availability of formalized libraries in machine learning theory, we aim to leverage these resources for dataset construction. A key challenge, however, lies in extracting subgoals from procedural-style proof scripts (Winograd, 1975; Wiedijk, 2012). Unlike declarative-style proofs (Syme, 1997), which progress by iteratively introducing and proving intermediate subgoals, procedural-style proofs instead transform the proof state step by step. As a result, existing goal-extraction tactics such as Lean 4's `extract_goal` can only capture the overall goal of a proof state, but not the finer-grained intermediate subgoals corresponding to each reasoning step. This limitation is particularly pressing as most proofs in current libraries are written in the procedural style. For example, in the proof of the lemma `linear_gradient` (see Figure 2 and Appendix C), procedural scripts such as `rw` and `simp` span nine lines, yet declarative statements (e.g., `have`) are entirely absent.

Fortunately, in human-written Lean 4 proofs, we observe a consistent pattern: *although proofs are expressed procedurally as sequences of tactics, each proof line typically corresponds to a single reasoning step identified by the human author.* In practice, human experts tend to first complete the reasoning informally and subsequently encode it in tactics. Accordingly, a single line frequently chains multiple tactics until the intended reasoning step has been fully realized, at which point a new line is introduced. Consider the running example in Figure 2, the expert combines two tactics (i.e., `repeat rw [dotProduct]` and `simp [mul_comm]`) within a single line to ensure that the resulting proof state aligns with the underlying informal reasoning. This observation motivates us to analyze human-written proofs at the line level.

Figure 2: An example of the `to_theorem` tactic illustrates its functionality. When applied to the tactic `repeat rw [dotProduct]; simp [mul_comm]`, it captures pre- and post-execution proof states, abstracts their transition, and synthesizes a subgoal.

We implement a customized Lean 4 tactic `to_theorem`, which automatically encodes line-level, procedural proofsteps into a new subgoal, with the corresponding proofsteps serving as their proofs. As shown in Figure 2, the tactic operates by recording the proof states immediately before and after executing the procedural steps `repeat rw [dotProduct]; simp [mul_comm]`. It then inserts the prior state as hypotheses and the subsequent state as the goal, thereby deriving a new theorem. This theorem effectively isolates a subgoal of the original proof, which can then be discharged directly by replaying the original procedural tactics.

The tactic `to_theorem` extracts the subgoal from a single line. We can naturally extend it to proof-segment extraction, where a sequence of tactics spanning multiple lines is first converted into the declarative format, and then its subgoal is extracted. For the running example `linear_gradient` in Figure 2, which totally comprises a nine-line proof, we can, in principle, extract up to nine single-line subgoals; four non-overlapping two-line subgoals; and so forth.

## 3.2 DATA CURATION

We categorize machine learning theories into two main types: optimization theory and probability theory. The former typically analyzes the convergence of optimization algorithms (e.g., gradient descent), while the latter addresses error bounds of machine learning models (e.g., Hoeffding's inequality). To develop FormalML, we expand and extract data based on two projects: **Optlib** (Li et al., 2024a) and **FoML** (Sonoda et al., 2025). For each project, we designate the top-level theorems (e.g., the convergence of gradient descent) for subgoal extraction, while preserving lower-level lemmas (e.g., Lipschitz continuity) as a local library for premises. We utilize the `to_theorem` tactic to extract the subgoals, and ultimately compile 4,937 theorems into FormalML.

Optlib (Li et al., 2024a; 2025a;b) is a project started in September 2023 that formalizes a wide range of optimization algorithms, including gradient descent (GD), subgradient method (SubGD), proximal gradient descent (PGD), Nesterov acceleration method (NAG), block coordinate descent (BCD), and alternating direction minimization method (ADMM). Building on Optlib, we extend and refine theorems related to these algorithms, and update some proofs to ensure compatibility.

FoML (Sonoda et al., 2025) is a recent project initiated in March 2025, primarily focused on formalizing the generalization error bound using Rademacher complexity. It also includes some important probability inequalities, such as the expectation inequality, the bounded differences inequality, and McDiarmid's inequality. Furthermore, we formalize Hoeffding's lemma, the Bennett inequality, and the Bernstein inequality into the library to enhance its comprehensiveness.

Each theorem in FormalML is stored in JSON format (See Appendix B), including:

(1) Source location metadata (file path and line number coordinates for theorem extraction).
(2) The formal Lean 4 theorem statement.
(3) Required module imports and namespace declarations.
(4) The complete tactic sequence constituting the original proof.
(5) Proof-relevant premises from the mathematical context.

Table 2: Statistics of theorems in FormalML across various machine learning theories.

| **Optimization** | GD | SubGD | PGD | NAG | BCD | ADMM | Other | Total |
|---|---|---|---|---|---|---|---|---|
| # Problems | 211 | 331 | 388 | 528 | 433 | 1,016 | 0 | 2,907 |
| **Probability** | Exp | Bennett | McDiarmid | Rademacher | Hoeffding | Measure | Other | Total |
| # Problems | 100 | 136 | 642 | 615 | 52 | 315 | 170 | 2,030 |

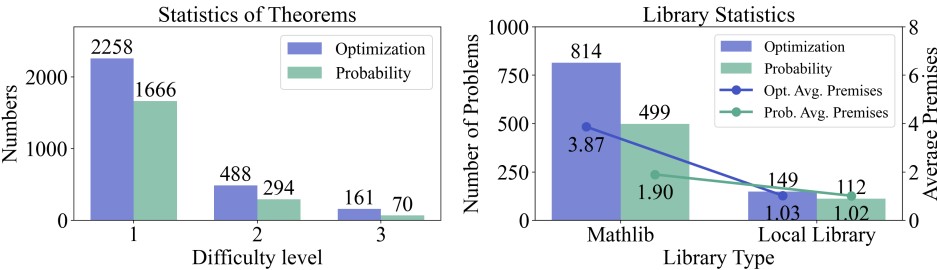

Figure 3: (Left) distribution by proving difficulty; (Right) statistics of premises.

This encapsulation ensures a complete proof context for each theorem, enabling independent evaluation. We release both the dataset and the evaluation code as open source.

### 3.3 BENCHMARK CHALLENGES

At first glance, subgoal completion may appear simpler than the proof generation for complex competition problems, in which LLM-based theorem provers have demonstrated their excellent performance. However, we identify that the proofstep generation task remains particularly challenging for LLM-based theorem provers, attributable to the following three key reasons.

- **Complex proof context.** In practice, users often rely on automated theorem provers to close subgoals within lengthy and complex proofs. This requires LLMs to comprehend all accumulated hypotheses and intricate goals to generate accurate proofs. However, current provers pay little attention to complex contextual reasoning.

- **Premise utilization.** Accurately using relevant premises from both local contexts and global libraries is essential, as many proofs rely on interdependent lemmas (Hu et al., 2024). A common workflow involves first performing premise selection via retrieval from a large library to form a small candidate set, which is then fed into an LLM for premise utilization. This paper focuses on the latter stage, as the ability of a model to choose and properly use premises from candidate sets is critical for practical subgoal completion. However, current LLM-based provers still perform inadequately in this regard (Yang et al., 2023), and few studies have systematically explored this.

- **Overthinking and efficiency.** Unlike competition problems that demand long chains of reasoning, subgoal completion is often repetitive and involves relatively straightforward reasoning. LLMs trained for complex competitions may *overthink* such tasks, exploring unnecessary inference paths that hinder performance (Chiang & Lee, 2024; Sui et al., 2025). Moreover, efficiency is crucial: it is impractical to expend substantial computational resources on proving subgoals that require only brief, direct proofs. Balancing reasoning depth with computational cost is thus a key challenge for practical subgoal completion.

### 3.4 BENCHMARK STATISTICS

**Summary.** We establish FormalML by systematically expanding and extracting theorems from two Lean 4 libraries: Optlib and FoML. The resulting benchmark comprises 4,937 unique theorems, each derived from a single proof step or proof segment within the top-level theorems presented in these libraries. For the extracted theorems, we categorize them according to their corresponding top-level theorems, as illustrated in Table 2. The results indicate that the benchmark encompasses a diverse range of theories in machine learning. The number of extracted theorems associated with each top-level theorem depends on the length of the corresponding proof. On average, hundreds of theorems can be successfully extracted from each top-level theorem.

**Proving difficulty.** To enable a fine-grained evaluation of LLM-based theorem provers, we categorize theorems based on their proving difficulty. Specifically, since the proofs in FormalML are mostly procedural style, we can use the proof length as a metric of proving difficulty (Zhang et al., 2024). We define three levels of proving difficulty, corresponding to proof lengths of 1, 3, and 5, respectively, and report the statistics of theorems across these levels in the left panel of Figure 3. In total, there are 3,924 theorems at difficulty level 1 and 1,013 theorems at higher difficulty levels.

**Premise utilization statistics.** We further examine the premise utilization difficulty within FormalML. In total, 1,547 theorems require the use of explicit premises. These theorems are divided into two categories based on the source of the premises: Mathlib and the local library. The right panel of Figure 3 reports the number of theorems in each category, as well as the average number of premises.

## 4 EXPERIMENTS

In this section, we conduct a comprehensive evaluation of state-of-the-art LLM-based theorem provers using our FormalML benchmark. Through systematic experiments, we identify several limitations and phenomena characterizing current theorem provers in practical subgoal completion tasks.

### 4.1 EVALUATING LLM-BASED THEOREM PROVERS ON FORMALML

**Theorem Provers**   We primarily focus on the two most effective types of LLM-based provers:

- **Best-First Tree-Search (BFS) Methods:** Each node in the search tree corresponds to a proof state, and a heuristic scoring function assigns priorities to nodes. The BFS algorithm is employed to explore the search space and derive the final proof. We evaluate two models in this framework: Reprover (Yang et al., 2023) and BFS-Prover (Xin et al., 2025).

- **Whole-Proof Generation Methods:** These types of models generate a complete formal proof in a single pass directly from the problem description, eliminating the need for additional state transitions or search procedures. We consider the following SOTA models: STP (Dong & Ma, 2025), Goedel-Prover (Lin et al., 2025a), Goedel-Prover-V2 (Lin et al., 2025b), Leanabell-Prover (Zhang et al., 2025), Kimina-Prover-Preview-7B (Wang et al., 2025), DeepSeek-Prover-V1.5 (Xin et al., 2024) and DeepSeek-Prover-V2 (Ren et al., 2025).

In addition, we compare several automation tactics in Lean 4, such as `simp`, `ring`, `linarith`, `tauto`, `grind`, and `aesop`. Lean 4's `aesop` (Limperg & From, 2023) is a strong and sophisticated, rule-based proof search engine. The ensemble means we combine all of the aforementioned automation tactics to try.

**Metrics & Evaluation**   We employ the Pass@$K$ metric to evaluate theorem provers, where $K$ represents the computational budget. This metric quantifies the proportion of problems for which at least one valid proof is discovered within the top $K$ generated attempts. For BFS methods, $K = E \times T$, where $E$ represents the number of tactics generated per expansion, and $T$ denotes the number of expansion iterations. Unlike prior studies that predominantly focus on competition-level theorem proving—characterized by highly challenging problems and lengthy proofs, thereby requiring large values of $K$, this work concentrates on the research level subgoal completion. Consequently, our benchmark emphasizes low computational budgets, corresponding to small values of $K$ ($\leq 8 \times 50$ for BFS methods and $\leq 32$ for whole-proof generation methods).

**Experimental Details**   For BFS methods, we implement and conduct interactive search based on the LeanDojo (Yang et al., 2023) framework. For whole-proof generation methods, we employed vLLM (Kwon et al., 2023) to generate proofs and evaluated their correctness using kimina-lean-server (Santos et al., 2025), which provides high-throughput Lean 4 verification. All experiments strictly adhered to the published parameter configurations and prompts for each model. All computations were performed on NVIDIA H800 GPUs.

**Main Results**   In Table 3, we show the performances of current theorem provers on our FormalML benchmark. Our experiments demonstrate that current whole-proof generation models have preliminarily acquired the capability to solve FormalML problems when increasing sample budgets, STP attaining the highest Pass@32 score of 63.21%. However, their Pass@1 performances remains at a relatively low level (peaking at merely 26.96%), falling short of the practical requirements for assisting mathematicians in completing Lean formal proofs. Notably, the BFS tree search strategy, despite consuming more computational resources, does not yield significant performance improvements

Table 3: Performance comparison of automation tactics and LLM-based theorem provers on FormalML. The best results are in bold, and the second-best are underlined.

| Method | Model Size | Sample Budget | Pass@$K$ (%) | | |
|---|---|---|---|---|---|
| | | | Optim. | Prob. | All |
| *Automation Tactics* | | | | | |
| simp | - | - | 19.85 | 15.96 | 18.25 |
| ring | - | - | 1.69 | 1.18 | 1.48 |
| linarith | - | - | 13.14 | 6.31 | 10.33 |
| tauto | - | - | 21.40 | 16.35 | 19.32 |
| grind | - | - | 15.69 | 11.28 | 13.87 |
| aesop (Limperg & From, 2023) | - | - | 45.24 | 40.49 | 43.29 |
| Ensemble | - | - | 51.53 | 44.58 | 48.67 |
| *Best-First Tree Search Methods* | | | | | |
| Reprover (Yang et al., 2023) | 229M | $4 \times 20$ | 21.80 | 16.15 | 19.48 |
| | | $8 \times 50$ | 27.82 | 20.44 | 24.79 |
| BFS-Prover (Xin et al., 2025) | 7B | $4 \times 20$ | 24.45 | 19.31 | 22.26 |
| | | $8 \times 50$ | 27.62 | 23.84 | 25.31 |
| *Whole-Proof Generation Methods* | | | | | |
| Kimina-Prover-Preview-7B (Wang et al., 2025) | 7B | 1 | 17.99 | 11.77 | 15.43 |
| | | 4 | 30.75 | 23.69 | 27.85 |
| | | 16 | 42.31 | 38.13 | 40.59 |
| | | 32 | 47.09 | 43.79 | 45.74 |
| Goedel-Prover (Lin et al., 2025a) | 7B | 1 | 8.22 | 13.50 | 10.39 |
| | | 4 | 21.32 | 28.97 | 24.47 |
| | | 16 | 37.29 | 41.28 | 38.93 |
| | | 32 | 44.34 | 46.11 | 45.07 |
| Goedel-Prover-V2-8B (Lin et al., 2025b) | 8B | 1 | 20.85 | 19.31 | 20.21 |
| | | 4 | 34.61 | 33.15 | 34.01 |
| | | 16 | 45.13 | 43.35 | 44.40 |
| | | 32 | 49.67 | 48.67 | 49.26 |
| Leanabell-Prover (Zhang et al., 2025) | 7B | 1 | 24.11 | 24.09 | 24.10 |
| | | 4 | 47.33 | 43.30 | 45.68 |
| | | 16 | 58.00 | 50.84 | 55.05 |
| | | 32 | 61.23 | 53.99 | 58.07 |
| STP (Dong & Ma, 2025) | 7B | 1 | 28.45 | 24.83 | 26.96 |
| | | 4 | 51.01 | 46.06 | 48.98 |
| | | 16 | 62.13 | 57.73 | 60.32 |
| | | 32 | **65.19** | **60.39** | **63.21** |
| DeepSeek-Prover-V1.5 (Xin et al., 2024) | 7B | 1 | 24.29 | 17.29 | 21.41 |
| | | 4 | 43.31 | 37.39 | 40.88 |
| | | 16 | 57.48 | 52.56 | 55.46 |
| | | 32 | 61.51 | 56.00 | 59.25 |
| DeepSeek-Prover-V2 (noCoT) (Ren et al., 2025) | 7B | 1 | 17.16 | 16.40 | 16.85 |
| | | 4 | 37.98 | 37.09 | 37.61 |
| | | 16 | 59.79 | 52.76 | 56.90 |
| | | 32 | 65.08 | 57.73 | 62.06 |
| DeepSeek-Prover-V2 (CoT) (Ren et al., 2025) | 7B | 1 | 16.23 | 21.58 | 18.43 |
| | | 4 | 25.73 | 32.56 | 28.54 |
| | | 16 | 33.54 | 39.36 | 35.93 |
| | | 32 | 37.39 | 42.51 | 39.50 |

(remaining below 30%). Additionally, most automation tactics generally underperform compared to LLMs. However, the powerful tactic aesop demonstrates strong performance, surpassing several LLMs under low sample budgets, though it still falls short of LLMs under high sample budgets. Even if all these tactics were integrated together, the ensemble method still not outperforms LLMs. This suggests that LLMs possess the potential to solve subgoals, but their costs need to be reduced. Furthermore, we observe that: 1) the performance ranking of models exhibits inconsistencies with

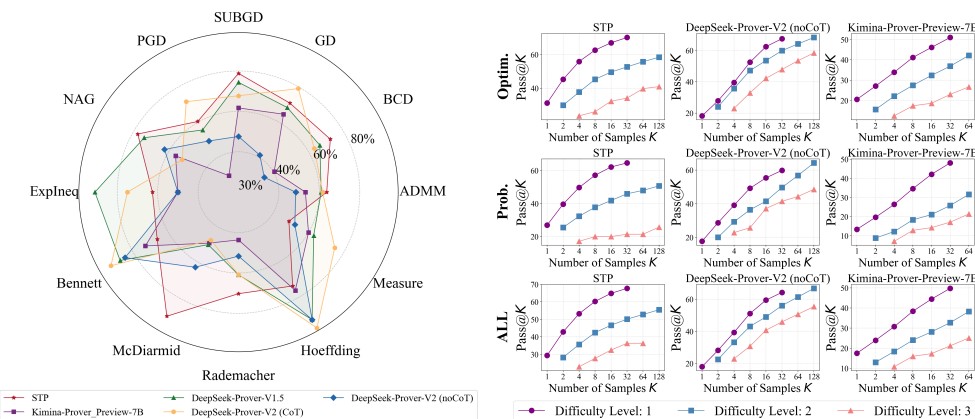

Figure 4: The left figure presents results of pass rate across various specific problem domains, while the right figure shows the performances under different difficulty levels.

the results from the miniF2F benchmark, suggesting that certain models exhibit varying degrees of overfitting to competition-level elementary theorem-proving tasks; 2) recent long-CoT provers (e.g. Kimina-Preview and Deepseek-Prover-V2) incorporating natural language assistance underperform on our FormalML benchmark (with pass@32 consistently below 50%), presenting a stark contrast to their strong performance on miniF2F.

**Performance Distribution**  We present the distributional results of performance across various specific problem domains (e.g., the proof of Hoeffding's inequality). The experimental results presented in left panel of Figure 4 demonstrate that the performance of existing models varies across different specialized areas. For instance, in the probability split, STP achieves significantly superior results on McDiarmid, far surpassing other methods. On Hoeffding, Deepseek-Prover-V2 (noCoT) delivers the best performance.

> **Finding 1:** Existing LLM-based theorem provers are inadequate as practical tools for assisting mathematicians in achieving subgoal completion *under low computational budgets*. Moreover, the performance of LLM-based theorem provers exhibits significant disparities across different specialized areas within FormalML.

## 4.2 EVALUATING PREMISE UTILIZATION ON FORMALML

While writing practical proofs, humans often need to apply appropriate premises from local theorem libraries or Mathlib. The experiments in this section are designed to evaluate the model's ability to utilize a given set of candidate premises through in-context learning to solve a proof subgoal. These experiments are carried out on a subset of our benchmark containing relevant premises, with the following specific design: for each proposition to be proved, we randomly sample candidate premises from a combined premise library comprising Mathlib and local premises, mixing in actually correct theorems to ultimately form a candidate set containing $M$ premises. All candidate premises are presented in a structured format within the model's context (the complete prompt template is provided in Appendix E). Considering the current context window limitations of LLM-based provers, this study employs four experimental settings ($M = 0$, $M = *$, $M = 10$, $M = 20$) for comparative analysis. Here, $M = 0$ denotes the baseline with no premises, while $M = *$ refers to the setting that includes only ground-truth premises.

Results are demonstrated in Table 4. We observe that nearly all models (except for Goedel-Prover-V2-8B) show improved performance when provided with accurate premises, indicating the benefit of supplying relevant contextual information. Although STP demonstrates strong performance across the full benchmark, it underperforms in premise utilization when $M = 10$ and $M = 20$. We attribute this to the absence of such data in its training set. We identify a representative case in Appendix G.1. In contrast, DeepSeek-Prover-V2 achieves robust results under both CoT and noCoT settings, with a high performance gain of approximately 10% . This indicates its strong premise utilization capability, which we believe stems from its training on extensive natural language reasoning data, thereby enhancing its general reasoning ability.

Table 4: Pass@16/32 (%) of premise utilization. Relative improvements compared to $M = 0$ are shown in green (increase) or red (decrease).

| Method | M = 0 | | M = * | | M = 10 | | M = 20 | |
|---|---|---|---|---|---|---|---|---|
| | 16 | 32 | 16 | 32 | 16 | 32 | 16 | 32 |
| STP | 54.56 | 57.72 | 57.85 (+3.29) | 61.86 (+4.14) | 52.23 (-2.33) | 56.43 (-1.29) | 52.17 (-2.39) | 56.04 (-1.68) |
| Goedel-Prover | 32.97 | 39.04 | 45.05 (+12.08) | 50.10 (+11.06) | 34.58 (+1.61) | 41.05 (+2.01) | 33.35 (+0.38) | 39.50 (+0.46) |
| Goedel-Prover-V2-8B | 45.24 | 51.26 | 45.18 (-0.06) | 50.29 (-0.97) | 45.05 (-0.19) | 49.90 (-1.36) | 46.09 (+0.85) | 50.36 (-0.90) |
| Leanabell-Prover | 46.67 | 50.42 | 49.32 (+2.65) | 53.65 (+3.23) | 47.45 (+0.78) | 50.87 (+0.45) | 46.93 (+0.26) | 50.29 (-0.13) |
| Kimina-Prover-Preview-7B | 34.71 | 39.56 | 35.16 (+0.45) | 40.14 (+0.58) | 34.65 (-0.06) | 39.69 (+0.13) | 35.49 (+0.78) | 39.75 (+0.19) |
| DeepSeek-Prover-V1.5 | 49.45 | 52.94 | 51.45 (+2.00) | 56.69 (+3.75) | 46.86 (-2.59) | 53.59 (+0.65) | 42.79 (-6.66) | 49.45 (-3.49) |
| DeepSeek-Prover-V2 (noCoT) | 53.46 | 58.37 | 59.47 (+6.01) | 71.86 (+13.49) | 58.82 (+5.36) | 69.29 (+10.92) | 57.40 (+3.94) | 65.80 (+7.43) |
| DeepSeek-Prover-V2 (CoT) | 34.39 | 37.94 | 44.36 (+9.97) | 47.81 (+9.87) | 44.34 (+9.95) | 47.71 (+9.77) | 42.73 (+8.34) | 46.74 (+8.80) |

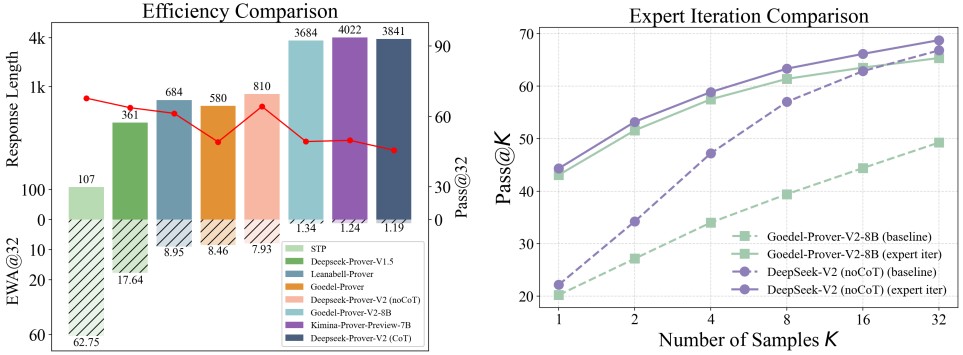

Figure 5: The left figure illustrates the efficiency comparison among current whole-proof generation provers, while the right shows performances before and after expert iteration.

> **Finding 2:** Existing LLM-based provers often struggle with premise utilization when candidate set is large, resulting in suboptimal performance. Moreover, even when provided with the ground-truth set of premises, most models appear to underutilize them, with the exception of DeepSeek-Prover-V2. This observation indicates that the primary performance bottleneck remains largely attributable to limitations of the base models.

## 4.3 EVALUATING FORMALML OF DIFFERENT DIFFICULTY LEVEL

We evaluate the performance of existing whole-proof generation models at varying difficulty levels on FormalML. A larger $K$ is used to assess problems of higher difficulty. As shown in right of Figure 4, experimental results indicate that model performance, both for the optimization and the probability split, decreases as difficulty increases. Notably, STP still achieves the highest results under difficulty levels 3 and 5, with Pass@128 scores of 55.5% and 33.36%, respectively. Results of other models are shown in Appendix F.2.

> **Finding 3:** Current LLM-based theorem provers tend to exhibit significantly degraded performance when handling higher-difficulty problems in FormalML.

## 4.4 EVALUATING THEOREM PROVER EFFICIENCY ON FORMALML

In formal theorem proving, efficient provers are crucial for practical applications. Recent studies demonstrate that RL-based long-CoT reasoning has been successfully integrated into systems such as DeepSeek-Prover-V2, Kimina-Prover, and Goedel-Prover-V2. These models employ deep natural-language reasoning before generating formal proofs. While they exhibit strong performance on competition-level mathematical problems, our experiments reveal no significant advantage on the FormalML benchmark (See Table 3). Moreover, these models generally suffer from low reasoning efficiency. To address the trade-off between reasoning efficiency and accuracy in FormalML scenarios, we use a novel evaluation metric—Efficiency-Weighted Accuracy (EWA@$K$)—defined as: EWA@$K$ = Pass@$K$ × $\frac{100}{\text{Response Length}}$. This metric balances performance and output efficiency by normalizing the proof success rate against the response length. To the right of Figure 5, we compare the pass rates and efficiency of eight whole-proof generation models on FormalML. The results show that the three long-COT models generate the highest average number of response tokens, far exceeding other models. However, their performance is inferior to others, resulting in the lowest

EWA@32 score. In contrast, STP achieves both the shortest output length and the highest pass rate on the FormalML benchmark, leading to the highest EWA@32 score.

> **Finding 4:** Long-CoT demonstrates no significant improvement in the pass rates of the subgoal completion task while simultaneously incurring substantial computational cost on FormalML.

### 4.5 EVALUATING FORMALML WITH EXPERT ITERATION

We further explored the performance of expert iteration on FormalML. We extracted 92,815 problems from five repositories (mathlib, PrimeNumberTheoremAnd, PFR, PhysLean, and scilean) using the `to_theorem` tactic, of which 88,174 were used for expert iteration training. For each problem, we generated 8 candidate proofs and sampled the correct ones for training. We performed 1 round of expert iteration on DeepSeek-Prover-V2 (noCoT) and Goedel-Prover-V2-8B. Training details are shown in Appendix D. Results are shown in the right panel of Figure 5. The experimental results indicate a substantial improvement in performance after expert iteration, especially in Pass@1. This suggests that expert iteration holds potential for enhancing subgoal completion tasks.

> **Finding 5:** It is demonstrated that expert iteration enhances performance on FormalML and shows potential for improving subgoal completion capabilities.

## 5 CONCLUSION

In this paper, we identify subgoal completion as a critical yet understudied task for helping mathematicians working on research-level problems. We introduce FormalML, the first benchmark specifically designed for this task in formal theorem proving. Using a Lean 4 translation tactic, we systematically extract 4,937 subgoal problems from foundational machine learning theory, covering optimization and probability. Evaluations of state-of-the-art LLM-based provers demonstrate persistent limitations in accuracy, efficiency, and premise utilization within complex proof contexts. Moreover, expert iteration has proven to be a promising direction for advancing subgoal completion. We anticipate that FormalML will catalyze the development of more capable provers to support subgoal completion.

## 6 ETHICS STATEMENT

This work complies with the ICLR Code of Ethics. It does not involve human subjects, sensitive data, or experiments with potential harm. All datasets are derived from publicly available formalized machine learning libraries in Lean4, with no proprietary information used. We believe the methods and results pose no ethical risks, and all authors take full responsibility for the integrity of the research.

## 7 REPRODUCIBILITY STATEMENT

We have made significant efforts to ensure the reproducibility of our results. The procedures for data curation and experiments are documented in the main text and supplementary materials. And we will release anonymous source code and scripts as supplementary material to facilitate reproduction of our experiments.

### ACKNOWLEDGEMENTS

This research was supported by the Key Program of Jiangsu Science Foundation (BG2024036, BK20243012), Leading-edge Technology Program of Jiangsu Science Foundation (BK20232003), Natural Science Foundation of China (62576162, 62025202), the Fundamental Research Funds for the Central Universities (022114380023), and the Frontier Technologies R&D Program of Jiangsu (BF2024059). Taolue Chen is partially supported by overseas grants from the State Key Laboratory of Novel Software Technology, Nanjing University (KFKT2023A04, KFKT2025A05) Yufeng Li (liyf@nju.edu.cn) and Xiaoxing Ma (xxm@nju.edu.cn) are the corresponding authors. Xiaowen Yang, Zihao Zhang, Jianuo Cao, Zhi Zhou, and Zenan Li are the core contributors.

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

APPENDIX

## A  THE USE OF LARGE LANGUAGE MODELS

In the course of this work, we made limited use of LLMs as auxiliary tools. Specifically, we used an LLM to assist in generating small utility Python scripts for tasks such as data preprocessing, formatting, and visualization. These scripts were only supportive in nature and did not contribute to the conceptual development, design of experiments, or the core scientific results of the paper. All key research ideas, theoretical analysis, experimental design, and writing of the main paper were carried out independently by the authors. The LLM was not used to generate or edit the scientific content of the manuscript, nor was it involved in producing research hypotheses or interpretations. The authors take full responsibility for the correctness, originality, and integrity of all content in the paper.

## B  DATA FORMAT SPECIFICATION

The FormalML dataset stores each theorem in a structured JSON format. Below is a detailed schema with field descriptions.

```
{
  "filename": string,
  "line": int,
  "tactic_state_before": string,
  "tactic": string,
  "tactic_state_after": string,
  "goal": string,
  "theorem_header": string,
  "formal_statement": string,
  "full_formal_statement": string,
  "retrieval": [
    {
      "library": string,
      "definition": string
    }
  ]
}
```

Based on the file `FoML/FoML/ForMathlib/Probability/Moments.lean` (Line 98), we further provide an illustrative example for some typical entries. In this line, the corresponding proof tactic is `apply aemeasurable_expt_hX`. By extracting the pre- and post-tactic states, we can formulate the corresponding subgoal as follows:

```
theorem extracted_formal_statement_27.{u_1}
{Ω : Type u_1}
{m : MeasurableSpace Ω}
{μ : Measure Ω}
[inst : IsFiniteMeasure μ]
(a b : ℝ)
{X : Ω → ℝ}
(hX : AEMeasurable X μ)
(h : ∀ᵐ (ω : Ω) ∂μ, X ω ∈ Set.Icc a b) :
let e := fun t ω => exp(t · Xω);
∀ (t' : ℝ), AEStronglyMeasurable (e t') μ := sorry
```

Regarding the retrieval entry, it records the library (e.g., `FoML`), and specifies the definition of the retrieved lemma. For example:

```
lemma aemeasurable_expt
{X : Ω → ℝ}
(t : ℝ)
(hX : AEMeasurable X μ) :
AEStronglyMeasurable (fun ω ↦ exp(t · X(ω))) μ
```

Table 5: Error type analysis for different theorem provers under pass@16 and pass@32.

| Setting | Model | has_error | is_valid_with_sorry | is_valid_no_sorry |
|---|---|---|---|---|
| Pass@16: Optimization | DeepSeek-Prover-V1.5 | 74.80 | 0.02 | 25.16 |
| | DeepSeek-Prover-V2 (CoT) | 63.16 | 10.68 | 15.49 |
| | Goedel-Prover | 90.08 | 0.00 | 9.92 |
| Pass@16: Probability | DeepSeek-Prover-V1.5 | 80.04 | 0.04 | 19.89 |
| | DeepSeek-V2 (CoT) | 61.94 | 7.65 | 22.77 |
| | Goedel-Prover | 85.20 | 0.00 | 14.80 |
| Pass@32: Optimization | DeepSeek-Prover-V1.5 | 75.09 | 0.02 | 24.87 |
| | DeepSeek-Prover-V2 (CoT) | 62.98 | 10.76 | 15.49 |
| | Goedel-Prover | 90.11 | 0.00 | 9.89 |
| Pass@32: Probability | DeepSeek-Prover-V1.5 | 79.88 | 0.04 | 20.04 |
| | DeepSeek-V2 (CoT) | 61.82 | 7.61 | 22.96 |
| | Goedel-Prover | 85.23 | 0.00 | 14.77 |

## C  THE FULL PROOF OF LEMMA `linear_gradient`

```
private lemma linear_gradient :
∀ x : (EuclideanSpace ℝ (Fin n)),
HasGradientAt (fun x : (EuclideanSpace ℝ (Fin n)) =>
(b •ᵥ (A *ᵥ x)))(Aᵀ *ᵥ b) x := by
  intro x
  rw [HasGradient_iff_Convergence_Point]
  intro a  apos
  use a ; use apos
  intro y _
  rw [dot_mul_eq_transpose_mul_dot,
    dot_mul_eq_transpose_mul_dot, ← dotProduct_sub]
  rw [EuclideanSpace.inner_eq_star_dotProduct]; simp
  repeat rw [dotProduct]; simp [mul_comm]
  apply mul_nonneg; linarith [apos]; apply norm_nonneg
```

## D  TRAINING DETAILS OF EXPERT ITERATION

Training used the AdamW optimizer with a learning rate of 1e-5, cosine learning rate schedule, warmup ratio of 0.1, batch size of 8 with gradient accumulation steps of 2 (effective batch size 16), and 3 epochs, with bf16 precision and DeepSeed ZeRO for distributed optimization. All experiments were conducted using GPU acceleration, and evaluation was carried out on the FormalML dataset.

## E  PROMPTS FOR PREMISE UTILIZATION

For premise utilization, we designed the following prompt, using DeepSeek-Prover-V2 as an illustrative example.

## F  ADDITIONAL RESULTS

### F.1  ERROR ANALYSIS

This section presents a detailed breakdown of error types from additional experiments conducted on three different theorem provers: DeepSeek-Prover-V1.5, DeepSeek-Prover-V2 (CoT), and Goedel-Prover. The results are categorized into three main types: has_error (Lean code with execution errors), is_valid_with_sorry (code that verifies but contains the sorry tactic), and is_valid_no_sorry (correctly verified proofs). These proportions were calculated for both pass@16 and pass@32 metrics across two different problem domains: Optimization and Probability. The tables below provide a comprehensive overview of these results, allowing for a direct comparison of the performance characteristics of each prover.

The results indicate that long-CoT models such as DeepSeek-V2-CoT tend to output "sorry" more frequently, but demonstrate lower rates of Lean errors. Conversely, Goedel-Prover seldom outputs "sorry" tactic but exhibit higher Lean error frequencies.

## F.2 RESULTS UNDER VARYING DIFFICULTY LEVELS

We supplement the experimental results for all eight whole-generation models across different difficulty levels, as presented in Figure 6 and Figure 7. The results demonstrate a statistically significant trend where all models exhibit performance degradation as the difficulty level increases. Among all evaluated models, DeepSeek-Prover-V2 demonstrates the most modest performance decline.

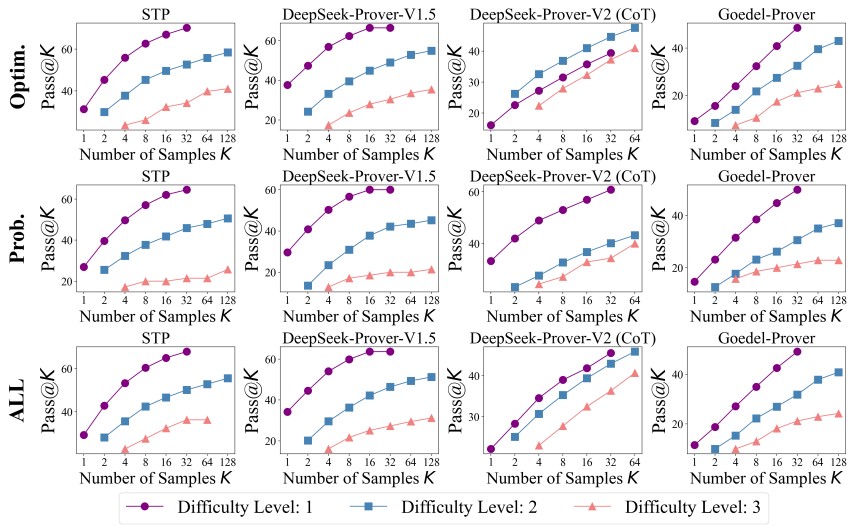

Figure 6: Results of pass rate under varying difficulty levels (Part 1).

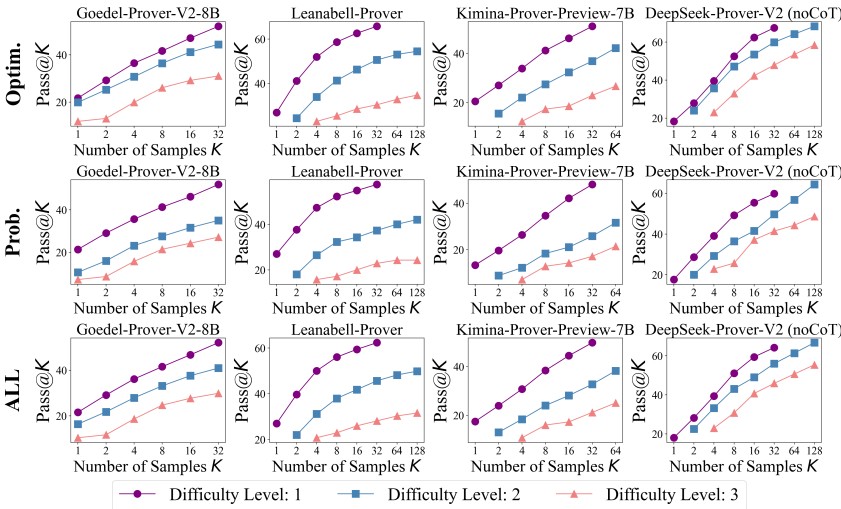

Figure 7: Results of pass rate under varying difficulty levels (Part 2).

> **The prompt for premise utilization of DeepSeek-Prover-V2 (CoT)**
>
> You can use some of the following lemmas or theorems: {all_definitions}
> Complete the following Lean 4 code:
> {formal_statement}
> Before producing the Lean 4 code to formally prove the given theorem, provide a detailed proof plan outlining the main proof steps and strategies.
> The plan should highlight key ideas, intermediate lemmas, and proof structures that will guide the construction of the final formal proof.

# G CASE STUDY

## G.1 CASES OF PREMISE UTILIZATION

We present a case of STP where the problem was initially solved without premise utilization, but errors emerged after candidate premises were provided.

> **Problem**
>
> ```
> theorem extracted_formal_statement_162
> {n m : ℕ+}
> {A : Matrix (Fin ↑m) (Fin ↑n) ℝ}
> {μ : ℝ}
> {μpos : 0 < μ}
> {Ane0 : A ≠ 0}
> (x : EuclideanSpace ℝ (Fin ↑n))
> (a : x ∈ Set.univ)
> (ε : ℝ)
> (εpos : ε > 0)
> (y : EuclideanSpace ℝ (Fin ↑n))
> (ydist : ‖x − y‖ ≤ ε / ↑↑n / μ)
> (h_original_goal : |μ * Σ x, |y x| − μ * Σ x_1, |x x_1‖ ≤ ε ) :
> |μ * Σ x_1, (|y x_1| − |x x_1|)| ≤ ε := sorry
> ```

> **Additional Theorems**
>
> ```
> theorem norm_one_proximal
> (lasso : h = fun y => μ · ‖y‖₁)
> (x : EuclideanSpace ℝ (Fin n))
> (xm : EuclideanSpace ℝ (Fin n))
> (tpos : 0 < t)
> (μpos : 0 < μ)
> (minpoint : ∀ i : Fin n, xm i = Real.sign (x i) * (max (abs (x i) − t * μ) 0)) :
> prox_prop (t · h) x xm
> theorem map_map
> (g : S →+* T)
> (f : R →+* S) :
> (s.map f).map g = s.map (g.comp f)
> theorem inner_add_left
> (x y z : F) :
> <<x + y, z>> =<<x, z>> + <<y, z>>
> theorem mul_pos
> {a b : Ordinal}
> (h₁ : 0 < a)
> (h₂ : 0 < b) :
> 0 < a * b
> lemma id :
> (fst φ).1.comp (inl φ) (add_neg_cancel 1) +
> (snd φ).comp (Cochain.ofHom (inr φ)) (add_zero 0) =
> Cochain.ofHom (1 _)
> theorem mul_sub
> (a b c : Ordinal) :
> a * (b − c) = a * b − a * c
> theorem id :
> (id V).NormNoninc
> ```

```
theorem smul_add
[Zero R]
[DistribSMul R V]
(x : HahnSeries Γ R)
(y z : HahnModule Γ" R V) :
x · (y + z) = x · y + x · z
theorem add_le_add_iff_right
{a b : Ordinal} :
∀ n : ℕ, a + n ≤ b + n ↔ a ≤ b
theorem add_right_cancel
{a b : Ordinal}
(n : ℕ) :
a + n = b + n ↔ a = b
theorem abs_abs
{x y : R}
(h : IsCoprime x y) :
IsCoprime |x| |y|
```

### Correct solution without premise utilization (STP)

```
import Mathlib
import Optlib

open Set Real Matrix Finset

open Set Real Matrix Finset NNReal
theorem extracted_formal_statement_162
{n m : ℕ+}
{A : Matrix (Fin ↑m) (Fin ↑n) ℝ}
{μ : ℝ}
{μpos : 0 < μ}
{Ane0 : A ≠ 0}
(x : EuclideanSpace ℝ (Fin ↑n))
(a : x ∈ Set.univ)
(ε : ℝ)
(εpos : ε > 0)
(y : EuclideanSpace ℝ (Fin ↑n)
(ydist : ‖x - y‖ ≤ ε / ↑↑n / μ)
(h_original_goal : |μ * Σ x, |y x| - μ * Σ x_1, |x x_1|| ≤ ε ) :
|μ * Σ x_1, (|y x_1| - |x x_1|)| ≤ ε := by
  cases' le_total (μ * Σ x_1, |y x_1|)
    (μ * Σ x_1, |x x_1|) with h h <;>
  simp_all [abs_of_nonneg, abs_of_nonpos, mul_add, mul_sub,
sub_add_eq_add_sub]
  <;> linarith
```

### Incorrect solution with premise utilization (STP)

```
import Mathlib
import Optlib

open Set Real Matrix Finset

open Set Real Matrix Finset NNReal

theorem extracted_formal_statement_162
{n m : ℕ+}
{A : Matrix (Fin ↑m) (Fin ↑n) ℝ}
{μ : ℝ}
{μpos : 0 < μ}
{Ane0 : A ≠ 0}
(x : EuclideanSpace ℝ (Fin ↑n))
```

```
(a : x ∈ Set.univ)
(ε : ℝ)
(εpos : ε > 0)
(y : EuclideanSpace ℝ (Fin ↑n))
(ydist : ‖x − y‖ ≤ ε / ↑↑n / μ)
(h_original_goal : |μ * Σ x, |y x| − μ * Σ x_1, |x x_1‖ ≤ ε) :
|μ * Σ x_1, (|y x_1| − |x x_1|)| ≤ ε := by

  have h₀ : (ε / ↑↑n / μ) * μ ≤ ε := by
    have h₁ : 0 < (↑↑n : ℝ) := by exact_mod_cast
      (Nat.pos_of_ne_zero (fun h ↦ by simp_all))
    have h₂ : 0 < μ := by exact_mod_cast μpos
    field_simp [h₁, h₂]
    <;> rw [div_le_iff (by positivity)] <;> nlinarith

  have h₁ : |μ * Σ x_1, (|y x_1| − |x x_1|)| ≤
  μ * Σ x_1, |y x_1 − x x_1| := by
    have h₂ : ∀ x_1, |μ * (|y x_1| − |x x_1|)| =
    μ * |y x_1 − x x_1| := by
      intro x_1
      rw [abs_mul, abs_of_pos (by exact_mod_cast μpos)]
    simp_all [abs_sum]

  have h₂ : μ * Σ x_1, |y x_1 − x x_1| ≤ ε := by
    calc
      μ * Σ x_1, |y x_1 − x x_1| ≤ μ * Σ x_1, ‖x − y‖ := by
        exact mul_le_mul_of_nonneg_left (by
          exact Finset.sum_le_sum (fun i _ ↦ abs_le_norm
          (y i − x i))) (by positivity)
      _ = (μ * Σ x_1 : Fin ↑n, ‖x − y‖) := by
        simp_all [Finset.sum_const]
      _ = (↑↑n * ‖x − y‖) * μ := by ring
      _ ≤ (↑↑n * (ε / ↑↑n / μ)) * μ := by gcongr
      _ = ε := by field_simp [npos, μpos] <;> ring

  linarith
  <;> simp_all
  <;> linarith
```

## G.2 CASES OF LONG-COT

We present the output results of the Long-CoT model on our benchmark. The model exhibits redundant responses and suboptimal performance, which stems from issues such as hallucination and overthinking. Specifically, the Long-CoT model frequently misjudges the complexity of given subgoals, resulting in the generation of excessively fine-grained subgoals (e.g., through the use of the have tactic). This tendency to overthink increases the risk of introducing erroneous statements, thereby undermining the validity of the output.

**Problem**

```
theorem extracted_formal_statement_1.{u_3, u_2, u_1}
{E₁ : Type u_1} {E₂ : Type u_2} {F : Type u_3}
[inst : NormedAddCommGroup E₁] [inst_1 : InnerProductSpace ℝ E₁]
[inst_2 : FiniteDimensional ℝ E₁] [inst_3 : NormedAddCommGroup E₂]
[inst_4 : InnerProductSpace ℝ E₂] [inst_5 : FiniteDimensional ℝ E₂]
[inst_6 : NormedAddCommGroup F] [inst_7 : InnerProductSpace ℝ F]
[inst_8 : FiniteDimensional ℝ F]
{admm : ADMM E₁ E₂ F} {admm_kkt : Existance_of_kkt admm}
(n : ℕ+) {xx : E₁} (a : xx ∈ univ) {yy : E₁} (a_1 : yy ∈ univ)
{aa bb : ℝ} (a_2 : 0 ≤ aa) (a_3 : 0 ≤ bb) (abh : aa + bb = 1)
(h_original_goal :
  aa · (OptProblem.A₁ E₂) xx + bb · (OptProblem.A₁ E₂) yy + (OptProblem.A₂ E₁) (ADMM.x₂ E₁ F
n.natPred) − OptProblem.b E₁ E₂ =
  aa · (OptProblem.A₁ E₂) xx + bb · (OptProblem.A₁ E₂) yy + (aa + bb) · ((OptProblem.A₂ E₁)
(ADMM.x₂ E₁ F n.natPred) − OptProblem.b E₁ E₂)) :
```

```
aa · (OptProblem.A₁ E₂) xx + bb · (OptProblem.A₁ E₂) yy + (OptProblem.A₂ E₁) (ADMM.x₂ E₁ F
n.natPred) − OptProblem.b E₁ E₂ =
  aa · (OptProblem.A₁ E₂) xx + bb · (OptProblem.A₁ E₂) yy + 1 · ((OptProblem.A₂ E₁) (ADMM.x₂ E₁ F
n.natPred) − OptProblem.b E₁ E₂) := by
```

## Model Response (Kimina-Prover-Preview-7B)

*(Abbreviated; ∼65 lines of repetitive reasoning and duplicate Lean code omitted from the original 83-line output.)*

### Internal Thinking

\# Solving the Formal Statement in Lean 4
I'm going to solve the problem of proving that [the goal equation] given the assumptions.
Let's think about this algebraically first. We have:
− `aa + bb = 1` (from the hypothesis `abh`)
− `0 <= aa` and `0 <= bb` (from `a_2` and `a_3`)
− The equation to prove: `x + y + (1 * z) = x + y + z` where z = (OptProblem.A2 . . .) − OptProblem.b . . .
This is just algebraic manipulation. Since `aa + bb = 1`, we have
  `(aa + bb) * z = 1 * z`. So the left side becomes `x + y + z`, which is the right side.
In Lean, we can prove this directly from h_original_goal. Let's complete the proof:
Now we have h_original_goal which states LHS = RHS. From algebra, (aa+bb)*z = 1*z.
−− ∼50 lines omitted: repeated explanation, substitution, linarith justification
The complete proof is remarkably concise.

### Proof

```
import Mathlib
import Optlib

open Set InnerProductSpace Topology Filter InnerProduct
open scoped Pointwise

theorem extracted_formal_statement_1.{u_3, u_2, u_1} {E₁ : Type u_1} {E₂ : Type
u_2} {F : Type u_3}
  [inst : NormedAddCommGroup E₁] [inst_1 : InnerProductSpace ℝ E₁] [inst_2 :
FiniteDimensional ℝ E₁]
  [inst_3 : NormedAddCommGroup E₂] [inst_4 : InnerProductSpace ℝ E₂] [inst_5 :
FiniteDimensional ℝ E₂]
  [inst_6 : NormedAddCommGroup F] [inst_7 : InnerProductSpace ℝ F] [inst_8 :
FiniteDimensional ℝ F]
  {admm : ADMM E₁ E₂ F} {admm_kkt : Existance_of_kkt admm} (n : ℕ+) {xx : E₁} (a : xx ∈
univ) {yy : E₁}
  (a_1 : yy ∈ univ) {aa bb : ℝ} (a_2 : 0 ≤ aa) (a_3 : 0 ≤ bb) (abh : aa + bb = 1)
  (h_original_goal :
    aa · (OptProblem.A₁ E₂) xx + bb · (OptProblem.A₁ E₂) yy + (OptProblem.A₂ E₁)
(ADMM.x₂ E₁ F n.natPred) −
      OptProblem.b E₁ E₂ =
    aa · (OptProblem.A₁ E₂) xx + bb · (OptProblem.A₁ E₂) yy +
    (aa + bb) · ((OptProblem.A₂ E₁) (ADMM.x₂ E₁ F n.natpred) − OptProblem.b E₁ E₂)) :
  aa · (OptProblem.A₁ E₂) xx + bb · (OptProblem.A₁ E₂) yy + (OptProblem.A₂ E₁) (ADMM.x₂
E₁ F n.natpred) −
      OptProblem.b E₁ E₂ =
    aa · (OptProblem.A₁ E₂) xx + bb · (OptProblem.A₁ E₂) yy +
    1 · ((OptProblem.A₂ E₁) (ADMM.x₂ E₁ F n.natpred) − OptProblem.b E₁ E₂) := by
  have h1 : aa + bb = 1 := abh
  linarith [h_original_goal, h1]
```

---

**Model Response (DeepSeek-Prover-V2 (CoT))**

*(Abbreviated; ∼195 lines omitted from the original 237-line output.)*

> **Internal Thinking**
>
> ### Detailed Proof and Analysis
> #### Understanding the Problem
> 1. The hypotheses `a : xx in univ` and `a_1 : yy in univ` are trivial (everything in univ).
> 2. The hypothesis `h_original_goal` is exactly the conclusion we need, under simplifying assumptions.
> 3. The goal: simplify `(aa + bb) * (...)` to `1 * (...)` using `abh : aa + bb = 1`.
>
> #### Simplifying the Goal
> `(aa + bb) * X = 1 * X` because `aa + bb = 1` by abh. Thus replace `(aa + bb) * ...` with `1 * ...`.
>
> #### The Proof
> 1. We know `aa + bb = 1` by abh.
> 2. Thus `(aa + bb) * X = 1 * X` (scalar mult linear).
> 3. Substitute into the original equation.
>
> #### Formal Proof Sketch
> 1. Replace `(aa + bb) * (...)` with `1 * (...)` using abh.
> 2. The result is exactly the RHS of the goal.
>
> #### Abstract Plan
> 1. Use abh to replace (aa+bb) with 1 in the scalar multiplication.
> 2. Simplify LHS to RHS: (aa+bb)*X = 1*X.
> 3. The rest follows trivially.
>
> ### Step−by−Step Abstract Plan
> 1. Simplify the scalar multiplication (aa+bb)*X = 1*X using aa+bb=1.
> 2. Substitute into h_original_goal.
> 3. Conclude: RHS of h_original_goal is the goal.
>
> ### Lean 4 `have` Statements
> theorem extracted_formal_statement_1.{...} : [...] := by
>   have h_main : (aa+bb)*X = 1*X := by sorry
>   have h_final : [goal] := by sorry
>   sorry
>
> ### Explanation
> h_main: scalar simplification. h_final: substitute h_main into h_original_goal. Final goal = h_final.

**Proof**

```
theorem extracted_formal_statement_1.{u_3, u_2, u_1} {E₁ : Type u_1} {E₂ : Type
u_2} {F : Type u_3}
  [inst : NormedAddCommGroup E₁] [inst_1 : InnerProductSpace ℝ E₁] [inst_2 :
FiniteDimensional ℝ E₁]
  [inst_3 : NormedAddCommGroup E₂] [inst_4 : InnerProductSpace ℝ E₂] [inst_5 :
FiniteDimensional ℝ E₂]
  [inst_6 : NormedAddCommGroup F] [inst_7 : InnerProductSpace ℝ F] [inst_8 :
FiniteDimensional ℝ F]
  {admm : ADMM E₁ E₂ F} {admm_kkt : Existance_of_kkt admm} (n : ℕ+) {xx : E₁} (a : xx ∈
univ) {yy : E₁}
  (a_1 : yy ∈ univ) {aa bb : ℝ} (a_2 : 0 ≤ aa) (a_3 : 0 ≤ bb) (abh : aa + bb = 1)
  (h_original_goal :
    aa · (OptProblem.A₁ E₂) xx + bb · (OptProblem.A₁ E₂) yy + (OptProblem.A₂ E₁)
(ADMM.x₂ E₁ F n.natPred) −
        OptProblem.b E₁ E₂ =
      aa · (OptProblem.A₁ E₂) xx + bb · (OptProblem.A₁ E₂) yy +
        (aa + bb) · ((OptProblem.A₂ E₁) (ADMM.x₂ E₁ F n.natPred) − OptProblem.b E₁ E₂)) :
  aa · (OptProblem.A₁ E₂) xx + bb · (OptProblem.A₁ E₂) yy + (OptProblem.A₂ E₁) (ADMM.x₂
E₁ F n.natPred) −
      OptProblem.b E₁ E₂ =
    aa · (OptProblem.A₁ E₂) xx + bb · (OptProblem.A₁ E₂) yy +
      1 · ((OptProblem.A₂ E₁) (ADMM.x₂ E₁ F n.natPred) − OptProblem.b E₁ E₂) := by
  have h_main : (aa + bb) · ((OptProblem.A₂ E₁) (ADMM.x₂ E₁ F n.natPred) − OptProblem.b
E₁ E₂) =
      1 · ((OptProblem.A₂ E₁) (ADMM.x₂ E₁ F n.natPred) − OptProblem.b E₁ E₂) := by
    have h₁ : aa + bb = 1 := abh
    rw [h₁]
    <;> simp [add_smul]
    <;> abel
  have h_final : aa · (OptProblem.A₁ E₂) xx + bb · (OptProblem.A₁ E₂) yy +
      (OptProblem.A₂ E₁) (ADMM.x₂ E₁ F n.natPred) −
      OptProblem.b E₁ E₂ = aa · (OptProblem.A₁ E₂) xx + bb · (OptProblem.A₁ E₂) yy +
      1 · ((OptProblem.A₂ E₁) (ADMM.x₂ E₁ F n.natPred) − OptProblem.b E₁ E₂) := by
    calc
      aa · (OptProblem.A₁ E₂) xx + bb · (OptProblem.A₁ E₂) yy + (OptProblem.A₂ E₁)
(ADMM.x₂ E₁ F n.natPred) − OptProblem.b E₁ E₂
        = aa · (OptProblem.A₁ E₂) xx + bb · (OptProblem.A₁ E₂) yy + (aa + bb) ·
((OptProblem.A₂ E₁) (ADMM.x₂ E₁ F n.natPred) − OptProblem.b E₁ E₂) := by
        rw [h_original_goal]; abel
      _ = aa · (OptProblem.A₁ E₂) xx + bb · (OptProblem.A₁ E₂) yy + 1 · ((OptProblem.A₂
E₁) (ADMM.x₂ E₁ F n.natPred) − OptProblem.b E₁ E₂) := by
        rw [h_main]; simp [add_smul]; abel
  exact h_final
```

# H  LIMITATIONS AND FUTURE WORK

Our current benchmark primarily focuses on a subset of main domains related to machine learning theory. In the future, we aim to expand into more relevant fields. Additionally, we plan to formalize certain aspects of ML theory using Lean, with the goal of encouraging more researchers to explore the use of LLM-based methods for assisting in formal theorem proving.

