# OpenReview forum: "FormalML: A Benchmark for Evaluating Formal Subgoal Completion in Machine Learning Theory"
_ICLR.cc/2026/Conference — ICLR 2026 Poster_

### Official Review · Reviewer_vdtg · 2025-10-31

**Soundness:** 3
**Presentation:** 3
**Contribution:** 3
**Rating:** 4
**Confidence:** 4

**Summary:**

This paper introduces a Lean 4 theorem proving benchmark, FormalML, with problems extracted from two formalization projects about machine learning theory. Unlike existing benchmarks, FormalML focuses on subgoal completion, rather than proving theorems from scratch -- motivated by the fact that users of interactive theorem provers are generally interested in calling automation in the middle of a partial proof. Experiments show that the benchmark is still generally challenging for models at the 7B scale especially at Pass@1, with STP doing best at 63% Pass@32. The paper shows several other analyses of adding premise retrieval (with mixed improvements) and expert iteration (generally improving model performance).

**Strengths:**

The paper shows a novel angle on dataset construction for theorem proving, focusing on subgoals. The extraction pipeline should be easy to use in other repositories beyond the two used here. It's good to see this expanding the evaluation of prover models beyond mathlib and including projects from the broader Lean community, which helps it represent more use cases of Lean. I'd expect this benchmark to be used by future work in LLMs for theorem proving, which currently has significant momentum.

The paper is generally very clear and describes the pipeline and the results well.

The results and analyses are also interesting, with techniques that are generally adopted now in LLM reasoning (like long CoT in recent reasoning models) not always being beneficial. Thus, this evaluation is complementary to existing benchmarks, not just showing the same patterns (besides being well motivated).

**Weaknesses:**

I have some issues with the evaluation and the interpretation of the results.

For evaluation, it would be helpful to know how many of the goals are solvable with simple automation tactics in Lean, such as grind, aesop, or canonical. Users generally try these (or similar, more domain-specific, eg ring) tactics first when they want proof automation. This would help readers get a better understanding of what fraction of the goals are indeed challenging for traditional automation.

For the interpretation, I think that actually some of the models seem to do rather well. Again, depending on whether these subgoals are out of reach of traditional automation, 25% pass@1 actually does not sound bad to me. Plus, STP with Pass@32 already achieves 63%. Thus, I suggest being less bold on the claim that they "are inadequate as practical tools" if the reference for this is this pass@1 or pass@4 evaluation.

Given the rapid progress in this area, and progressive gains in efficiency, it's likely that smaller models on smaller budgets might achieve similar performance on this benchmark in not too long. While FormalML might be useful to track that progress, there's a risk of it being saturated quickly. It would help to perhaps include other projects that are at the moment significantly more challenging, to motivate the community to work on those. If the best results on level 3 goals (in Figure 4) are already over 60%, it's seems that the current methods are perhaps on track to tackle them with smaller budgets/models given sufficient efficiency improvements on all fronts.

For the evaluation of retrieval, it seems that there's no model doing retrieval itself, and the experiment is mainly testing the ability of models to use in-context statements. While this is a useful analysis, the descripiton is a bit misleading. For instance, the conclusion is that models "often struggle with premise retrieval". But retrieval here is done manually, not by the model, so the model is not struggling with retrieval, but rather in using the premises. Thus, here I suggest simply rephrasing the conclusion.

**Questions:**

- What is the performance of simple automation tactics on FormalML?
- Did you analyze the examples of theorems solved by DeepSeekProver without CoT that failed with CoT? What seem to be the most common failure modes with CoT when that happens (ie when it succeeds without it)?
- In the retrieval experiment, how do models perform in a "perfect retrieval" scenario, where you give them exactly (and only) the premises used in the proof? That would give a sort of ceiling for the impact of retrieval here.
- Was there any reason to not include other repositories (such as those used in the Expert Iteration experiment) in the benchmark? It seems to me that the pipeline should be very generally applicable.

---

> ### Author Response · Authors · 2025-11-21
>
> We thank Reviewer vdtg for their insightful feedback.  Your suggestions are very actionable.
>
> We address your concerns below:
>
> 1.  **On Weakness 1 & Question 1 (Simple Automation Tactic Baselines):**
> This is an excellent and critical suggestion.  In our revised version, we execute standard automation tactics (e.g., simp, linarith, ring, tauto, grind, and aesop) on FormalML and incorporate the results into our evaluation (Table 3).  Since our benchmark is built on Lean v4.18, the recently introduced Canonical tactic is not supported.  These results indicate that, with the exception of aesop, all other methods perform significantly worse than LLMs.  Although aesop achieves a 43% success rate, it still falls short of high-budget LLMs.
>
> 2. **On Weakness 2 (Interpretation of Results):**
> This is a fair point about our tone.   We revise our claims.   We rephrase to emphasize that  high budget (Pass@32) results (like 63% for STP) show promise and the low budget performance highlights persistent limitations and the need for further research.
>
> 3.  **On Weakness 3 (Saturation Risk):**
> This is a valid concern.    We argue that the benchmark is far from saturated, as even the top-performing model achieves only 63% Pass@32 (high-budget conditions).    This indicates that, when employing LLMs as human assistants, approximately 40% of subgoals remain uncompleted in high-budget scenarios.    We anticipate that a suitably designed LLM-based human-in-the-loop formal co-pilot should achieve 90%+ subgoal completion even under low-budget settings, which would substantially alleviate the human burden in proof formalization.
>
> 4.  **On Weakness 4 and Question 3(Retrieval Interpretation):**
> You are absolutely correct.  We have revised the paper to change "premise retrieval" to "premise utilization" to more accurately describe our experiments.  We have provided the results of the "perfect retrieval" scenario (M=*) in Table 4.  We find that this setting will bring significant benefits to LLMs, indicating the necessity of providing accurate premises.  Thank you for your advice.
>
> 5.  **On Question 2 (CoT Failure Analysis):**
> In Appendix G.2, we have provided an example of DeepSeek-V2-CoT.   Specifically, the Long-CoT model frequently misjudges the complexity of given subgoals, resulting in the generation of excessively fine-grained subgoals (e.g., through the use of the have tactic).   This tendency to overthink increases the risk of introducing erroneous statements, thereby undermining the validity of the output.   More generally, we find that the model exhibits redundant responses and suboptimal performance, which stems from hallucination and overthinking.
>
> 6.  **On Question 4 (Other Repos):**
> We believe our framework is general.  We selected ML theory as our initial testbed precisely because it represents a complex research area where formalization is hindered by numbers of mechanical subgoal details.  Moreover, inclusion of additional repositories would significantly increase the volume of test data (we have already ~5k data) which could hinder testing efficiency for both our team and the broader research community.  However, our framework remains suitable for constructing large training data, as demonstrated in our paper in Section 4.5.
>
> Thank you for these concrete and helpful suggestions.

---

> > ### Comment · Reviewer_vdtg · 2025-11-25
> >
> > Thank you for engaging with the comments.
> >
> > I find it interesting that even in the M=* scenario, models don't improve much overall. Thus, I think that finding 2 can be strengthened - in general it seems that models are not able to use even the ground truth set of premises that well (except for DSP-V2). This suggests that the main bottleneck for now is still largely in the base models, not in the retrieval per se, and I think that result is useful for the community to know about.
> >
> > As one last suggestion: in Table 3, it would be good to report results for the ensemble of the automation tactics, rather than just for each of them individually (since users typically try several of them to see if any of them suffices, before investing much time thinking). If you have the data for which theorems were proved by which tactic, this should be easy to compute without re-running the system.
> >
> > I think the new results and revisions to the text have made the paper stronger, and I revised my score to support it.

---

> > > ### Author Response · Authors · 2025-11-27
> > >
> > > **Thank you very much for your positive feedback and constructive suggestions.** These comments have been instrumental in significantly improving the quality of our paper. We have thoroughly revised the paper based on your suggestions.
> > >
> > >
> > > Specifically, regarding **Finding 2**, we have strengthened our discussion to emphasize that even when provided with the ground-truth set of premises, most models appear to underutilize them, hoping to provide a meaningful insight for the community.
> > >
> > > Furthermore, we have updated **Table 3** to include the results for the ensemble of all automation tactics. The overall performance is 48.67%, which is approximately 5 percentage higher than pure Aesop but remains inferior to the LLM-based methods.

---

### Official Review · Reviewer_dUjc · 2025-11-01

**Soundness:** 3
**Presentation:** 3
**Contribution:** 3
**Rating:** 4
**Confidence:** 4

**Summary:**

This paper proposes **FormalML**, a Lean 4-based benchmark for evaluating LLM performance in *subgoal completion*—a task designed to align with the vision of "mathematicians providing high-level proof frameworks while models verify intermediate conclusions" (as illustrated in Figure 1). The benchmark extracts 4,937 subgoals from optimization and probability theories using the custom `to_theorem` tactic, which decomposes procedural proofs into declarative subgoals. Evaluations of state-of-the-art theorem provers (e.g., STP, DeepSeek-Prover-V2) reveal underwhelming performance, though this is partially attributed to the subgoals’ nature: most can be solved with basic tactics like `rw` and `simp`, making them "out-of-distribution" for models not trained on such granular steps.

**Strengths:**

1. **Vision Alignment with Practical Research Needs**: The paper identifies subgoal completion as a critical gap in LLM-aided formal theorem proving, directly targeting the real-world workflow where mathematicians prioritize "inspiration and intuition" (high-level frameworks) over tedious technical details. This vision addresses a tangible pain point in mathematical research and formal verification.
2. **Novel Benchmark for a Underexplored Task**: FormalML fills a void in existing benchmarks by focusing on subgoal completion (rather than full-proof generation) and integrating premise retrieval and research-grade ML theory contexts. It provides a structured, quantifiable foundation for evaluating LLM capabilities in intermediate proof steps.
3. **Transparent Analysis of Model Limitations**: The paper acknowledges that poor model performance is partially due to the "out-of-distribution" nature of the subgoals (solvable via basic tactics like `rw`/`simp`). This honest attribution avoids overstating limitations and frames performance gaps as actionable, context-dependent challenges.

**Weaknesses:**

While the paper’s core vision of human-AI collaboration in formal theorem proving is compelling, the implementation of subgoal extraction via `to_theorem` is criticized for being overly simplistic and failing to fully realize Figure 1’s intended workflow. A key unresolved question is how training on these subgoals impacts models’ performance on standard full-proof generation tasks.

1. **Overly Simplistic Subgoal Extraction via `to_theorem`**: The `to_theorem` tactic decomposes procedural proofs into subgoals tied to "single-step tactic state transitions," which fails to capture the nuance of Figure 1’s vision. These subgoals are not the "non-trivial intermediate conclusions" mathematicians would need help with—instead, they are granular, tactic-dependent steps that lack the complexity of real-world research gaps.
2. **Limited Justification for Subgoal Utility**: Since most extracted subgoals can be solved with basic Lean tactics (e.g., `rw`, `simp`), the paper does not fully explain why LLMs are needed for this task. The subgoals do not address the "hard technical gaps" that would truly liberate mathematicians, reducing the practical impact of the benchmark.
3. **Incomplete Analysis of Training Transfer**: The paper evaluates model performance on subgoal completion but does not explore how training on these simplistic subgoals affects performance on standard full-proof generation tasks. This omission leaves a critical gap in understanding the tradeoffs of subgoal-focused training.

**Questions:**

1. The `to_theorem` tactic extracts subgoals from existing procedural proofs, but these subgoals are far simpler than the "non-trivial intermediate conclusions" in Figure 1. Do the authors have plans to refine subgoal extraction (e.g., integrating human annotations to mark meaningful research gaps) to better align with the paper’s core vision?
2. Most subgoals can be solved with basic tactics like `rw` and `simp`, raising questions about their utility. How do the authors plan to evolve FormalML to include more complex, tactic-agnostic subgoals that reflect the "hard technical details" mathematicians actually struggle with?
3. The paper notes that poor model performance is partially due to "out-of-distribution" subgoals. Have the authors tested whether training models explicitly on these `to_theorem`-extracted subgoals improves performance—and if so, does this training cause performance drops (i.e., "catastrophic forgetting") on standard full-proof generation tasks?
4. Given that basic tactics already solve most subgoals, what concrete advantages does an LLM-based approach offer for subgoal completion? For example, can LLMs handle edge cases or premise combinations that basic tactics cannot, and if so, how is this validated in the benchmark?

---

> ### Author Response · Authors · 2025-11-21
>
> We thank Reviewer dUjc for their feedback.
>
> We believe there may be a slight misunderstanding regarding the precise scope of our objective, which we wish to respectfully clarify: Our primary goal is to position the LLM as a "detail-completion assistant" in the mathematical formalization process. We are not focused on having the LLM solve high-difficulty theorems like IMO-level problems. Instead, we are concerned with aiding humans in the completion of simple, mechanical, yet strictly formalized subgoals that arise during formal proof construction (e.g., certain inequality derivations that come naturally in human prose but still require detailed formal proof, as illustrated in Figure 1). Our aim is to liberate mathematicians from tedious technical details so they can focus on high-level proof sketch.
>
> Here are our responses to your main concerns and questions:
>
> **Regarding Simplicity and Utility of Subgoals (Weakness 1 2 & Questions 1 2, 4)**
>
> 1. First, we would like to politely consult the reviewer: We are somewhat confused by the term “tactic-agnostic subgoals,” as all existing studies on LLM-assisted theorem proving have focused on tactic-based modes in Lean or similar systems. We have not encountered similar expressions in relevant literature. **Could you kindly elaborate on the precise meaning of “tactic-agnostic”?** Based on our assumption, you might be referring to subgoals that require complex reasoning and a combination of multiple tactics, rather than a single simple one. Our benchmark already includes a significant proportion of such complex subgoals.
>
> 2. You suggest that most subgoals can be solved using basic tactics (like rw and simp), questioning the necessity of LLMs. We understand this concern, but our experimental evidence clearly demonstrates that the task is non-trivial:
>   - Our experiments on FormalML show that the automated simp tactic achieves a performance of only 18.24%. This is significantly lower than state-of-the-art LLM models. This result strongly confirms that **basic automated strategies cannot solve the majority of our subgoals**, thus validating the non-trivial nature of the task and the utility of LLMs.
>   - Even a seemingly simple rw tactic in our benchmark often requires the precise application of a complex chain of premises and theorems. For instance, the rw step in the linear_gradient proof example (from the main text) involves rw a long string of theorems like 'rw [dot_mul_eq_transpose_mul_dot, dot_mul_eq_transpose_mul_dot, ← dotProduct_sub]'. The exact application of these multiple theorems is a major challenge for non-LLM or current LLM-based provers and is a key area where human mathematicians invest significant, yet tedious, time in searching and reasoning.
>
> 3. The core advantage of the LLM approach is its ability to:
>   - Accurately utilize candidate premises.
>   - Apply complex combinations of tactics and theorem chains.
> This capability is precisely where we expect the LLM to function: handling the tedious, detailed proofs that human mathematicians prefer to delegate, thereby substantially reducing the formalization burden.
>
> **Regarding Training Transfer Analysis (Weakness 3 & Question 3)**
> In fact, our paper does not claim that poor model performance is partially due to "out-of-distribution" subgoals. Since we are focusing on the subgoal completion task, we do not expect the model to be capable of full proof generation, nor is it necessary. We argue that even if such targeted training may lead to a decline in the model’s performance on full proof generation tasks such as MiniF2F, it is still justified as long as it significantly enhances its practical utility in real-world formalization workflows (i.e., subgoal completion), as also confirmed by the performance improvement reported in Section 4.5.

---

### Official Review · Reviewer_mSD1 · 2025-11-01

**Soundness:** 3
**Presentation:** 2
**Contribution:** 3
**Rating:** 6
**Confidence:** 4

**Summary:**

The paper introduces FormalML, a Lean 4 benchmark targeting subgoal completion: filling in short but nontrivial proof steps within research-level ML theory proofs rather than producing full proofs end-to-end. The authors implement a Lean tactic, `to_theorem`, that extracts line- or segment-level subgoals from procedural proof scripts, and curate 4,937 problems spanning optimization and probability (with premise retrieval needs and complex contexts). A broad evaluation of recent LLM-based provers shows modest Pass@1, steep drops with difficulty, mixed gains from premise retrieval, and poor efficiency for long-CoT methods. They also show expert iteration improves performance, especially at low budgets.

**Strengths:**

1. Novelly shifts focus from full-proof generation (competition-style) to research-adjacent subgoal completion, a practical interface between human sketches and formal verification. Benchmarks combine premise retrieval with complex research contexts (from Optlib/FoML), which most prior Lean benchmarks avoid.

2. Introduces a symbolic extraction tactic (`to_theorem`) that converts procedural Lean steps into declarative subgoals at line/segment granularity, novel and broadly useful for dataset creation and tooling.

3. Clear, reproducible dataset construction pipeline. Thorough experiments across tree-search and whole-proof families, multiple budgets, domain splits, and difficulty levels; plus targeted retrieval experiments and an efficiency metric. Additionally includes error analysis (sorry vs. Lean errors) and a concrete ablation on expert iteration showing meaningful gains.

4. Provides a large and focused ML-theory benchmark likely to steer future model/training design (e.g., retrieval-aware training, efficiency-aware decoding).

**Weaknesses:**

1. Difficulty is proxied by proof length. That can correlate but isn’t identical to semantic difficulty. It would be good to add alternative signals and/or compare with human annotations.

2. I would go more carefully about the claim that long-CoT "shows no benefit". It seems supported on FormalML but may be dataset-specific. Also, long-CoT prompts also prime models for full-proof search, not subgoal snippets. More work would be needed before concluding broadly on this.

3. Authors note both single-line and multi-line segment extraction. It would be good to show how segment length affects success, retrieval load, and Pass@K. This would inform optimal chunking for future datasets.

4. Authors evaluate models some of which may have seen Optlib/FoML/mathlib during pretraining. How did you guard against data leakage? Any checks that extracted subgoals aren’t near-duplicates of public training proofs?

**Questions:**

1. Concentration on ML theory (Optlib + FoML) may limit generality to other formal domains (algebraic topology, number theory, verification). Consider adding a small cross-domain diagnostic split (e.g., mathlib analysis/algebra lemmas) to evaluate transfer and prevent domain-overfitting.

2. Consider adding refinement-at-most-N-steps interactive baselines (tool-use or repair loops) to gauge whether light interactivity beats one-shot generation.

3. Table 3 lists “Goedel-Prover-V2-8B” with size "7B". Can you double check this is consistent?

4. Slightly inconsistent terminologies: both "Mcdiarmid" and "McDiarmid" appear; both "Godel" and "Goedel" are spelt.

---

> ### Author Response · Authors · 2025-11-21
>
> We thank Reviewer mSD1 for their thorough review and for recognizing the novelty of the subgoal completion task and our to_theorem tactic. We appreciate the constructive feedback and will address the weaknesses and questions.
>
> 1. **On Weakness 1 (Difficulty Proxy):**
> We acknowledge that proof length is not a perfect measure of semantic complexity. However, we employed line count as a proxy for difficulty based on the observation that each proof line written by human typically corresponds to a single reasoning step. Consequently, longer proofs generally imply a greater number of requisite logical steps, serving as an indirect indicator of difficulty. Furthermore, our empirical results (Figures 4, 6, and 7) demonstrate a consistent decline in model performance as proof length increases, suggesting that this metric effectively approximates semantic complexity. While rigorous human annotation would be ideal, it was not feasible given the scale of our benchmark (approximately 5,000 problems).
>
> 2. **On Weakness 2 (Long-CoT Claim):**
> You are right. We rephrase our conclusion to be more precise, acknowledging this may not generalize to all subgoal tasks and we will do further research about this observation in the future.
> 3. **On Weakness 3 (Segment Length):**
> As the premise utilization experiments (we have changed premise retrieval to premise utilization in the paper) does not incorporate difficulty in this split, it is not suitable for comparative analysis of proof length. In addition, we conducted experiments to evaluate how segment length (i.e., the difficulty level) influences Success and Pass@K. The results are presented in Figure 4, Figure 6, and Figure 7, and they may not have been noticed.
>
> 4. **On Weakness 4 (Data Leakage):**
> Firstly, the formal libraries to construct our benchmark are up-to-date (as of April 2025), with some theorems and proofs supplemented by ourselves, after the release of the most neural theorem provers in the paper. Secondly, our to_theorem tactic generates new, declarative subgoal problems from the original procedural proofs. These specific subgoal statements do not exist in the source files. Therefore, the target problems themselves are entirely novel to the evaluated models.
>
> 5. **On Questions 1 (Cross-Domain Evaluation):**
> In Section 4.5, we constructed a training set from mathlib using the to_theorem tactic to evaluate the FormalML benchmark. This training set covers a wide range of fundamental mathematical domains (analysis, algebra, etc.) and exhibits a clear domain difference from the benchmark used for testing. The experimental results demonstrate that a model trained on mathlib can significantly improve the performance on FormalML, which we consider can be a cross-domain evaluation.
>
> 6. **On Question 2 (Interactive methods):**
> We agree that evaluating interactive or repair-based baselines represents a valuable direction for future work.  However, we have some reservations: interactive methods often entail high computational costs, which could conflict with the emphasis on efficiency in subgoal completion.  Moreover, incorporating such interactive components would introduce an additional dimension that may not directly reflect the core capability of LLM-based provers in subgoal completion, and could slightly diverge from the primary focus of this paper.  We plan to explore this direction in subsequent research.
>
> 7. **On Typos (Q3, Q4):**
> Thank you for catching these! We immediately correct the "7B" size in Table 3 and standardize the spellings of "McDiarmid" and "Goedel" throughout the paper.
>
> Thank you for the detailed feedback, which will significantly improve our paper's clarity and rigor.

---

> > ### Comment · Reviewer_mSD1 · 2025-11-25
> >
> > Thanks for the responses! I would like to maintain my positive evaluation of this work, and further raise the sub-score for Presentation given the changes authors proposed. Thank you for the good work!

---

> > > ### Author Response · Authors · 2025-11-25
> > >
> > > Thank you for your positive feedback and for raising the presentation score. We are delighted to hear that you are satisfied with our revisions.

---

### Official Review · Reviewer_7LrB · 2025-11-06

**Soundness:** 3
**Presentation:** 3
**Contribution:** 3
**Rating:** 8
**Confidence:** 4

**Summary:**

This paper introduces FormalML, a benchmark of 4937 problems extracted from machine learning theory in a subgoal completion context (filling in sorries). The authors also use expert iteration to significantly improve the abilities of two open-source models.

**Strengths:**

- The paper curates a large and interesting benchmark for theorem proving in Lean sourced from existing and reputable data sources.
- Almost all leading theorem-proving models are included and at multiple sampling budgets.
- The authors analyze current models from various perspectives: low computational budgets, premise retrieval, and long CoT.
- The authors show the effectiveness of expert iteration on their benchmark.

**Weaknesses:**

- The paper is limited to the domain of machine learning theory, which is somewhat narrow. Other than ease of data sourcing, there is no apparent reason why this needs to be the case. It would be great to expand the work to other domains as well.
- Relatedly, while the findings are interesting, the authors do not comment on how the benchmark can inform and advance downstream research in formal theorem proving, especially in other domains.
- As stated, the premise retrieval part of the paper does not consider embedding-style retrieval and only completion-based retrieval with the argument that using the same model enhances efficiency. However, a small embedding model could be cheap and produce better results.

**Questions:**

- The gains from expert iteration are very drastic for Goedel-Prover-8B and DeepSeek-V2, but the two models exhibit different patterns across sample budget. It is surprising that Goedel-Prover is showing uniform gains across all sample budgets, while DS-V2 shows very large gains for pass@1 and little gain for pass@32. Do you have any insights into this?
- How do the insights from this benchmark transfer to theorem proving in general? Do you envision the target to be a model specialized for machine learning theory proving?

---

> ### Author Response · Authors · 2025-11-21
>
> We sincerely thank Reviewer 7LrB for their positive feedback and insightful comments.
>
> We address your concerns below:
> 1. **On Weaknesses 1 & 2 and Question 2:**
> You are correct that FormalML is currently focused on ML theory. We selected this domain as our initial testbed precisely because it represents a complex research area where formalization is hindered by numbers of mechanical subgoal details.
>  However, we strongly believe our framework and findings are general:
>   - The task of subgoal completion is universal to any mathematician using a formal prover.
>   - The to_theorem tactic we developed is general and can be applied to any Lean project to extract subgoals.
>   - The insights (e.g. high cost of long CoT,  challenges in complex context and premise utilization) are likely to be fundamental properties of current LLM-based models, not specific to ML theory.
> We have revised the paper to better position FormalML as an in-depth case study for this general task, and we plan to extend the benchmark to other domains in future work. We are working for developing a general subgoal completion model and ensuring its applicability within this benchmark.
>
> 2. **On Weakness 3 (Retrieval):**
> We apologize for the lack of clarity in our initial wording.   We have revised the paper to change "premise retrieval" to "premise utilization" to more accurately describe our experiments.
> Our goal was not to evaluate the retrieval step itself (e.g., how to find the premises from a large library).   Instead, our focus was to evaluate: "Given a small set of candidate premises (which we guarantee contains the correct ones), how well can the model utilize these premises via in-context learning to solve the subgoal?"
> We believe this setup is valuable because it isolates the model's reasoning ability from its retrieval ability.   This allows our benchmark to seamlessly connect with existing and future retrieval systems.   A powerful embedding-based retriever (as you suggested) could be paired with a model that performs well on our benchmark to create a highly effective system.
>
> 3. **On Question 1 (Expert Iteration Gains):**
> This is an excellent observation.  We think that the inherent differences in the base models' capabilities likely account for the different relative gains. DeepSeek-V2's pass@k curve shows rapid growth at smaller k values, while Goedel-Prover's curve increases roughly linearly with log(k).  Despite the difference in the magnitude of improvement, we note that their final performance curves tend to consistent after training.  This consistency is likely due to both models using the same training data.
>
> Thank you again for your valuable feedback, which will help us strengthen the paper.

---

> > ### Comment · Reviewer_7LrB · 2025-11-21
> >
> > Thanks for the response! I maintain my score and support for the acceptance for this paper.

---

> > > ### Author Response · Authors · 2025-11-22
> > >
> > > Thank you again for your time, support, and valuable advice on improving our work.

---

### Author Response · Authors · 2025-11-21
**Revision&Rebuttal Summary**

We sincerely thank all reviewers  for their insightful comments and constructive feedback. Your suggestions have been valuable in helping us improve the paper. Below, we provide a summary of the key revisions made to our paper:

1. **We replace “premise retrieval” to “premise utilization”** to better reflect our focus. Rather than evaluating the retrieval process, we aim to assess how well the model can use a small set of candidate premises to solve subgoals via in-context learning. This setup isolates the model's reasoning ability from its retrieval performance.

2. **We include a set of automation tactics as baseline comparisons** in Table 3, providing a clear perspective on the challenge level of FormalML and the current capabilities of LLMs. The results show that most automation tactics still fall short, and while Aesop outperforms LLMs under a low inference budget, it remains inferior to high-budget LLMs (e.g., pass@32). This suggests that LLMs hold promising potential for subgoal completion tasks.

3. We correct typos and imprecise statements throughout the paper.

**After Rebuttal**:

**Three of the four reviewers (vdtg, mSD1, 7LrB) have expressed support for the paper or raised their scores.**
The remaining reviewer (dUjc, initial score 4) has not responded to our reply. This reviewer’s primary concern was that the task might be too simple and solvable with basic tactics. Our supplementary experiments (where simp achieved only 18.24% success) effectively address this point. We also note that this reviewer seems to have misunderstood our subgoal completion task and we reclarified this in our response.

---

### Meta-Review · Area_Chair_HHkM · 2026-01-10

**Summary:**

-	This paper introduces FormalLM, a benchmark comprising 4,937 extracted subgoal completion cases from machine learning theory proofs. The authors employ expert iteration on two open-source models to demonstrate performance improvements. Additionally, they analyze the benchmark's utility for theorem proving through a comprehensive evaluation of recent LLM-based provers, examining techniques including limited-budget proving, premise retrieval, and long chain-of-thought reasoning.
-	Following the review process, several key concerns were raised:
-	Reviewer 7LrB questioned the benchmark's limited scope to the machine learning domain and its potential applicability to downstream research. Also, about the oversimplified definition of completion-based retrieval tasks.
-	Reviewer mSD1 raised multiple concerns: (1) the appropriateness of using proof length as a difficulty metric, (2) missing analysis regarding segment length effects, (3) potential data leakage issues, (4) the generalizability of this benchmark on theory proof, and (5) typographical errors throughout the manuscript.
-	Reviewer dUjc argued that the proposed task is too simplistic to provide meaningful challenges.
-	Reviewer vdtg provided detailed questions regarding the tactical approaches employed and identified potential biases in the definition of the retrieval methodology.

During the rebuttal process, the authors addressed the above concerns, clarified potential biases, and made corresponding revisions to the manuscript.
In summary, this work provides a valuable benchmark for the subgoal completion problem in machine learning theory formalization. While its generalizability to diverse downstream tasks remains limited, it offers meaningful contributions to the development of formal theorem proving and theoretical understanding.
After careful consideration, I recommend acceptance of this paper.

**Reviewer Concerns:**

The main concerns raised by all reviewers were appropriately addressed in the authors' response. While Reviewer dUjc did not participate in the discussion phase, their primary concerns were nevertheless addressed in the rebuttal.

**Reviewer Scores:**

•  Reviewer 7LrB: Maintained all original scores

•  Reviewer mSD1: Increased representation score to 3; all other scores unchanged

•  Reviewer dUjc: Maintained all original scores

•  Reviewer vdtg: Increased overall rating to 7; all other scores unchanged

---

### Decision · Program_Chairs · 2026-01-26

Accept (Poster)